# Regional medical inter-institutional cooperation in medical provider network constructed using patient claims data from Japan

**Yu Ohki** [1]*, **Yuichi Ikeda** [1]*, **Susumu Kunisawa**[2], **Yuichi Imanaka**[2]

**1** Graduate School of Advanced Integrated Studies in Human Survivability, Kyoto University, Kyoto, Japan, **2** Graduate School of Medicine, Kyoto University, Kyoto, Japan

* ohki.yu.65a@st.kyoto-u.ac.jp (YO); ikeda.yuichi.2w@kyoto-u.ac.jp (YI)

**Data Availability Statement:** The data generated and analyzed during this study cannot be shared publicly, due to the Ethical Guidelines for Medical and Biological Research Involving Human Subjects

## Abstract

The aging world population requires a sustainable and high-quality healthcare system. To examine the efficiency of medical cooperation, medical provider and physician networks were constructed using patient claims data. Previous studies have shown that these networks contain information on medical cooperation. However, the usage patterns of multiple medical providers in a series of medical services have not been considered. In addition, these studies used only general network features to represent medical cooperation, but their expressive ability was low. To overcome these limitations, we analyzed the medical provider network to examine its overall contribution to the quality of healthcare provided by cooperation between medical providers in a series of medical services. This study focused on: i) the method of feature extraction from the network, ii) incorporation of the usage pattern of medical providers, and iii) expressive ability of the statistical model. Femoral neck fractures were selected as the target disease. To build the medical provider networks, we analyzed the patient claims data from a single prefecture in Japan between January 1, 2014 and December 31, 2019. We considered four types of models. Models 1 and 2 use node strength and linear regression, with Model 2 also incorporating patient age as an input. Models 3 and 4 use feature representation by node2vec with linear regression and regression tree ensemble, a machine learning method. The results showed that medical providers with higher levels of cooperation reduce the duration of hospital stay. The overall contribution of the medical cooperation to the duration of hospital stay extracted from the medical provider network using node2vec is approximately 20%, which is approximately 20 times higher than the model using strength.

## Introduction

As the global population ages, there is an increasing need to establish healthcare systems that can sustainably provide high-quality healthcare to the elderly. According to the United

established jointly by Japanese ministries. However, other researchers may send data access requests to Office of Research Promotion, General Affairs and Planning Division, Kyoto University (E-mail: 060kensui@mail2.adm.kyoto-u.ac.jp; Tel: +81-75-753-9301).

**Funding:** This work was supported by JSPS KAKENHI (Grant Number: JP19H01075) from the Japan Society for the Promotion of Science and Health and Labour Sciences Research Grant from the Ministry of Health, Labour and Welfare, Japan (Grant Number: 21IA1005 and 21FA1012). YO thanks the establishment of university fellowships towards the creation of science technology innovation by JST (Grant Number JPMJFS2123). Y. Ikeda and YO would also like to acknowledge Ripple, which is providing financial support through its University Blockchain Research Initiative. The funders had no role in study design, data collection and analysis, decision to publish, or preparation of the manuscript.

**Competing interests:** The authors have declared that no competing interests exist.

Nations' World Population Prospects 2019, the aging rate, which is the percentage of the total population aged 65 and above, is predicted to increase from 9.3% in 2020 to 15.9% in 2050 [1]. We expect to emerge as a super-aged society on a global scale. We need to provide efficient medical services to address this situation.

Care coordination with multiple participants can improve medical outcomes while limiting healthcare costs [2]. We used network science to evaluate the cooperation of medical providers and physicians to provide efficient healthcare. Researchers have proposed various methods to construct physician and medical provider networks to share or transfer patients [3]. Previous studies on physician networks have examined the relationship between network features and medical cost [4–9], quality of care [10], and mortality [11] as a patient-level medical outcome measure and the relationship between various medical outcomes and network features at the regional level [12]. Physician networks contain information about medical cooperation and the quality of healthcare.

Compared with studies on physician networks, few reports of studies on medical provider networks exist. The extent to which medical cooperation, extracted from the medical provider network, affects the quality of health care provision is an interesting question. Ostovari and Yu and Gander et al. examined the relationship between the network features of medical providers and the patient-level medical outcome [13, 14]. These studies assigned one medical provider spending most resources on each patient and did not incorporate the usage patterns of multiple medical providers in a patient's medical care series. These studies, including those of physician networks, have used general network features to represent cooperation among medical providers and physicians. However, the general network features extract only one aspect of cooperation, which may not fully represent the information. We expect the network to contain more information on medical cooperation to explain the quality of healthcare provision.

The healthcare system in Japan requires more extended hospital stays than those in other countries. In Japan, the acute, recovery, and chronic care hospitals are separated to share healthcare provision and efficiently provide healthcare with the aim of shortening the duration of hospital stay. We focus on this regional medical inter-institutional cooperation in Japan. From this perspective, it is crucial to examine the contribution of cooperation among medical providers to a patient's usage pattern of medical providers. We consider that smooth cooperation among medical providers leads to good results. However, it is unclear how much cooperation takes place, how cooperation is implemented, and what its implications are for individual patients. Therefore, it is necessary to establish the relationship between medical cooperation and the quality of healthcare, considering the usage patterns of medical providers.

We analyzed the medical provider network to examine the overall contribution to healthcare quality by cooperation among the medical providers. This study focused on: i) the method of feature extraction from the network; ii) incorporating the usage pattern of medical providers; iii) expressive ability for statistical models. First, we extracted features representing each medical provider in a network structure through machine learning. Second, we integrated these features of medical providers used by a patient with each case and obtained features corresponding to the usage patterns of medical providers. Third, we used a machine learning model with a high expressive ability to explain the healthcare quality based on these features. Additionally, we selected femoral neck fractures as the target disease. Femoral neck fractures frequently occur in the elderly. The treatment phase shifts from hospitalization in the acute phase to rehabilitation in the recovery phase and long-term care after discharge [15–17]. Therefore, this ailment is suitable for evaluating the effect of regional medical inter-institutional cooperation in the transition of treatment phases.

We analyzed the claims data of patients residing in a prefecture in Japan between January 1, 2014 and December 31, 2019 to construct a network representing the medical providers'

cooperation. We consider several models: a model using the node strength and linear regression to a model using feature representation by node2vec and regression tree ensemble. We used the regression models to examine the relationship between the duration of hospital stay and features of nodes in the medical provider network. We also analyzed the relationship between medical cooperation and network structure using community analysis.

This study clarifies how the strength of each medical provider relates to the hospital stay. However, the statistical relationship between strength and the duration of hospital stay is weak, even when age is considered. In contrast, the model using the feature representation by node2vec, input variables, and a regression tree ensemble as a regression model was more explanatory than the model using strengths. The community analysis revealed that the high-strength network structure has short distance, high clustering, and weak disassortability. This result indicates that a horizontally decentralized network structure is more effective for medical cooperation than a centralized network.

This study is valuable because we reveal that regional medical inter-institutional cooperation represented by the medical provider network is an important factor in shortening the hospital stay. Its overall contribution to the duration of hospital stay for femoral neck fracture is approximately 20%. In addition, we investigated effective network structure for medical cooperation. These findings suggest ways to function the regional medical inter-institutional cooperation. This study contributes to an efficient healthcare system to cope with a super-aged society.

The remainder of this paper is organized as follows. The following section describes the role of medical cooperation among healthcare providers for a femoral neck fracture. We then describe the data, data analysis method, statistical model, and community analysis in the "Data and Method" section. The "Results" section presents the relationship between the features of medical providers and the duration of hospital stay based on the constructed network. Using a community analysis, we also examine the relationship between the network structure and medical cooperation. The "Discussion" section discusses the results, and the "Summary" section summarizes and concludes this paper.

## Regional medical inter-institutional cooperation

Regional medical inter-institutional cooperation is one of the characteristics of the Japanese healthcare system. The conceptual diagram of the regional medical inter-institutional cooperation for femoral neck fracture is shown in Fig 1. We consider that the medical provider network represents the cooperation among medical providers.

Patients with femoral neck fractures usually undergo surgery in an acute phase hospital after the fracture, then transition to rehabilitation in a recovery phase hospital, and they receive care at home through outpatient care hospital or at a nursing home after discharge [15–17]. Quality of surgery and rehabilitation in acute and recovery phase hospitals affects recovery. The quality of long-term care in community hospitals and nursing homes also affects recovery. We expect that effective cooperation among these medical providers will improve the quality of care. In addition, we must consider the social factors that maintain people's health status. The impact of various social services on health was investigated in [18]. Patients with femoral neck fractures require social service support before and after the fracture.

We examine the factors associated with mortality [19–23], recovery [24], and quality of life [25] for hip fractures, including the femoral neck fractures. As described below, each factor is related to one of the actors presented in Fig 1, such as the patients, medical providers (acute phase hospitals, recovery phase hospitals, and community hospitals), nursing homes, or administration. Firstly, there are patient-specific factors such as age and sex. The older the

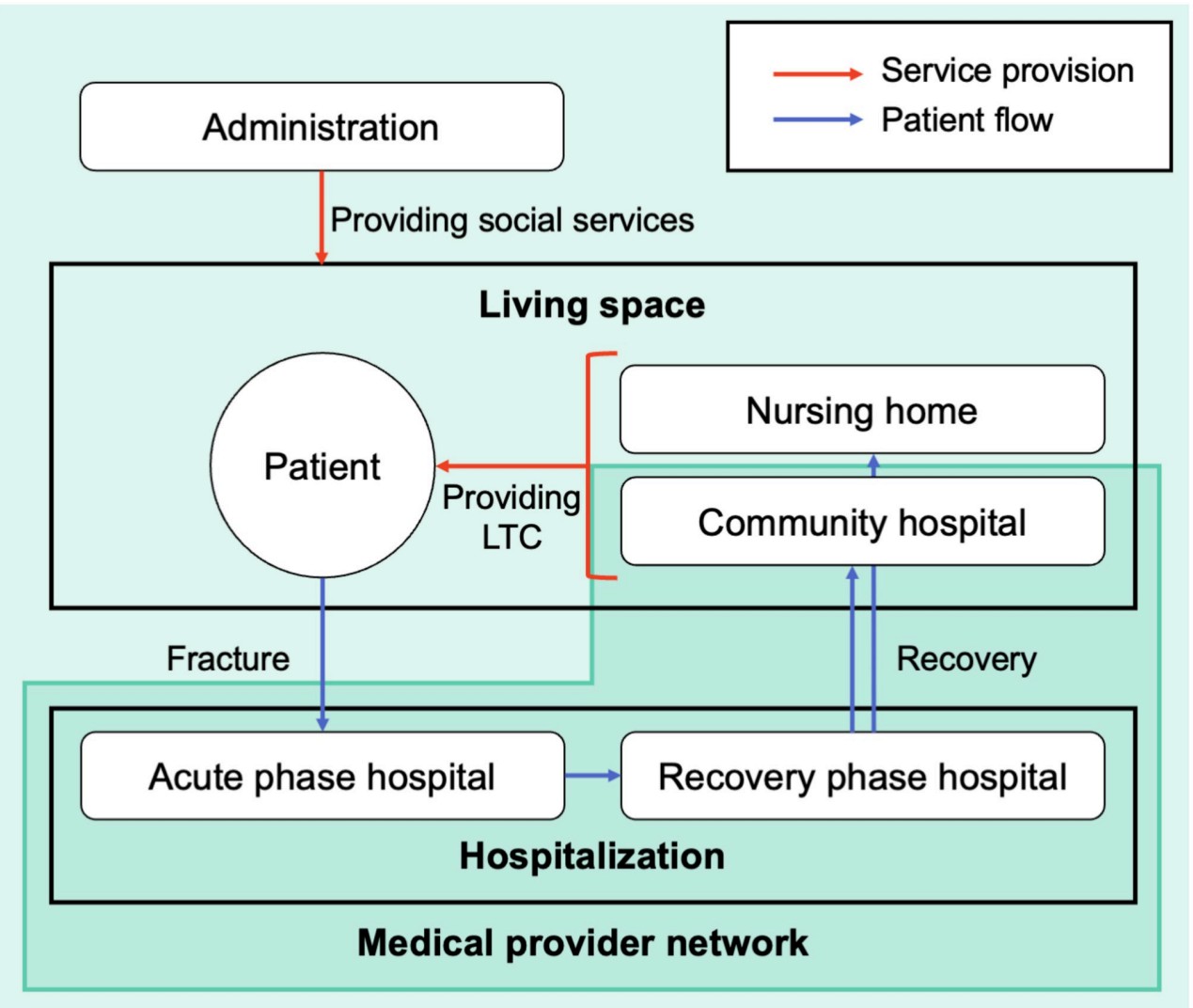

**Fig 1. Conceptual diagram of regional inter-institutional medical cooperation for treating femoral neck fracture.** The red arrows represent the flow of patients. The blue arrows represent the service provision relationship. Patients with femoral neck fractures are hospitalized in the acute and recovery phase hospitals. Community hospitals and nursing homes provide long-term care (LTC) after discharge. The administration provides social services to maintain and improve the health of patients before and after a fracture.

patient, the higher was the risk of a femoral neck fracture. Men are more likely to die than women, and women are more likely to recover than men. Secondly, health factors were related to the patients' physical condition and health status. Low weight and nutritional status and diseases, such as renal and cardiovascular diseases, may increase the risk of fractures in patients. Thirdly, there are environmental factors such as residential conditions. Patients residing in nursing homes were more likely to die than those residing at home. Fourthly, there are social service factors. Social services improve patient health and the environment. Fifthly, there are medical care factors, such as the quality of surgery and rehabilitation. Sixthly, there are medical cooperation factors. We consider the time from fracture to surgery and the duration of hospital stay as medical care cooperation factors because effective cooperation can reduce them.

Based on the above, the quality of healthcare provided to patients $Q$ is determined by the following factors: patient-specific factor $P$, health factor $H$, Environmental factor $E$, social

service factor $S$, medical care factor $M$, and medical cooperation factor $C$. Thus, we obtain the following functional:

$$Q = f(P, H, E, S, M, C). \tag{1}$$

This study examines the contribution of medical cooperation factor $C$ from analysis of medical provider networks.

## Data and methods

We analyzed the patient claims data, constructed a medical provider network, and calculated the duration of hospital stay to model the relationship between medical inter-institutional cooperation and the quality of healthcare provision. We considered statistical models to examine the relationship. We also used community analysis to examine the relationship between medical cooperation and network structure.

### Data

We analyzed anonymized personal-level patient claims data for residents of a prefecture in Japan. The data included inpatient and outpatient records related to the femoral neck fractures from January 1, 2014, to December 31, 2019. This study included 9,496 patients and 863 medical providers, respectively in Table 1. Approximately 94% of the patients used medical providers within the prefecture. The data items for inpatient claims include patient ID, medical provider ID, and dates of admission, discharge, and surgery. The data items for outpatient claims include patient and medical provider ID and the visit date.

### Medical provider network

We constructed a medical provider network using the patient claims data. At first, we constructed a patient–medical provider bipartite graph. The $ij$−component $B_{ij}$ of the bi-adjacency matrix $B(N_p \times N_m)$ corresponding to the bipartite graph represents inpatient or outpatient care of medical provider $i$ to patient $j$. $N_p$ and $N_m$ denote the number of patients and medical providers, respectively. Also when patient $j$ uses medical provider $i$ multiple times, $B_{ij} = 1$. Column vector $B_i$ in the $i$th row of the bipartite graph is a vector representing the patients who use the medical provider $i$.

We then project this bipartite graph onto the medical provider network. As the self-loop is removed, the $ij$−component $A_{ij}$ of the adjacency matrix $A(N_p \times N_p)$ is as follows:

$$A_{ij} = \begin{cases} w_{ij} & (i \neq j) \\ 0 & (i = j) \end{cases}. \tag{2}$$

The weight $w_{ij}$ between medical providers $i$ and $j$ is cosine similarity between vectors $B_i$ and $B_j$ that represent patients using medical providers $i$ and $j$, respectively. It is an undirected

**Table 1. Summary of patient claims data.**

|  | Number of records | Number of patients | Number of medical provider | Percentage of medical records in the prefecture |
|---|---|---|---|---|
| Inpatient claims data | 13193 | 8775 | 411 | 94.2% |
| Outpatient claims data | 153035 | 7187 | 692 | 93.8% |
| Total | 166228 | 9496 | 863 | 93.8% |

As there is an overlap in the number of patients and medical providers in the inpatient and outpatient claims data, the total values do not equal the sum of both.

graph from the definition of weights, such as the following equation:

$$w_{ij} = w_{ji} = \frac{\boldsymbol{B}_i \boldsymbol{B}_j^T}{\sqrt{\sum_{i=1}^{N_p} B_{ij}^2} \sqrt{\sum_{j=1}^{N_p} B_{ij}^2}}.$$ (3)

In many previous studies, the weights were the number of patients shared among medical providers: $w_{ij} = \boldsymbol{B}_i \boldsymbol{B}_j^T$. However, we used cosine similarity for the weights to normalize the effect of the size of medical providers.

## Network features

We calculated the network features representing the structural characteristics of the medical provider network. We calculated the average degree, strength, distance, clustering coefficient, and assortativity.

The degree $k_i$ is the number of edges of node $i$. Average degree $\langle k \rangle$ is the mean of the degrees of all nodes: $\langle k \rangle = \Sigma_i k_i / N = 2L/N$, where $N$ denotes the number of nodes, and $L$ denote the edges. $\langle k \rangle$ represents the average number of medical providers with which a medical provider shares patients.

The strength $s_i$ is the sum of the weights of the edges of node $i$. Average strength $\langle s \rangle$ is the mean of the strength of all nodes: $\langle s \rangle = \Sigma_i s_i / N$. The weight $w_{ij}$ is normalized for the effect of the size of the medical provider, as defined in Eq (3). We consider the value of strength as the level of cooperation of each medical provider with neighboring medical providers.

Distance $d_{ij}$ is the number of edges when connecting the nodes $i$ and $j$ with the shortest path length in the network. The average distance $\langle d \rangle$ is the mean of the distances between all nodes: $\langle d \rangle = \Sigma_{i,j;i \neq j} d_{ij} / N(N-1)$.

The cluster coefficient $C_i$ is the fraction of edges between neighboring nodes of node $i$: $C = 2l_i / k_i(k_i - 1)$, where $l_i$ is the number of edges between neighboring nodes of $i$. Average clustering coefficient $\langle C \rangle = \Sigma_i C_i / N$.

Assortativity $r$ is the Pearson correlation coefficient of the degree of the nodes at both ends of the edge, $r = \Sigma_{ij}(k_i k_j - \langle k \rangle^2) / \Sigma_i (k_i - \langle k \rangle)^2$. Networks with $r > 0$ represent degree assortativity networks, and networks with $r < 0$ represent disassortativity networks.

## Statistical model

The medical provider network represents cooperation among medical providers. We consider a statistical model with the features extracted from this network as input variables and the quality of healthcare provision as an output variable. We use strength and feature representation by node2vec network embedding as input variables to extract information about medical cooperation from the medical provider network. Strength is an explicit feature of cooperation. However, we obtain feature representations by embedding the network structure in the feature space. Duration of hospital stay was used as the output variable.

**Duration of hospital stay.** We calculated the duration of hospital stay $D^{in}$ using the patient claims data. Firstly, we extracted a series of medical records from inpatient and outpatient claims data, as shown in Fig 2(a). We set two types of thresholds, $\tau_1$ and $\tau_2$, for the extraction. The threshold of hospital transfer $\tau_1$, determines the inpatient transfer. If a patient is admitted to another hospital within $\tau_1$ days of the date of discharge from one hospital, the patient transfer is related to the same fracture. Threshold of care continuity $\tau_2$ was used to integrate the pre-and post-hospitalization outpatient records in a series of medical records. We considered an outpatient record within $\tau_2$ days of the date of admission, discharge, or visit as outpatient care related to the same fracture. We extracted a series of medical records related to

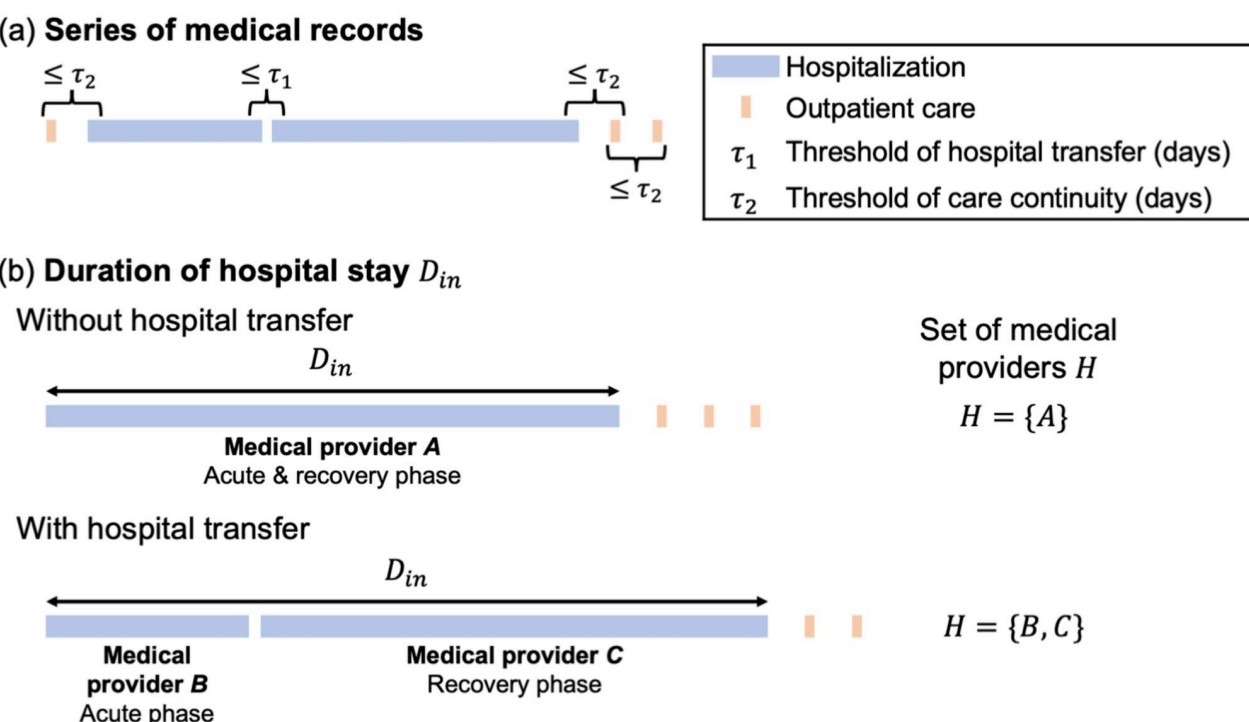

**Fig 2. Methods for calculating duration of hospital stay $D^{in}$.** (a) Extracting a series of medical records using two types of threshold $\tau_1$ and $\tau_2$. The threshold of hospital transfer $\tau_1$ determines the inpatient transfer. The threshold of care continuity $\tau_2$ is used to integrate pre-and post-hospitalization outpatient records into the series of medical records. (b) Calculating the duration of hospital stay $D^{in}$ from each series of medical records. In the case of medical records with hospital transfers, the date of admission at the first medical provider to the date of discharge at the last medical provider is $D^{in}$. $H$ denotes set of medical providers where a patient is hospitalized.

a single fracture in a patient using $\tau_1$ and $\tau_2$. We set $\tau_1 = 10$ d and $\tau_2 = 35$ d. We determined these thresholds to be the inpatient and outpatient intervals distributions, as shown in S1 Fig.

Secondly, we calculated the duration of hospital stay $D^{in}$ in each series of medical records. A series of medical records may or may not include hospital transfers. Without hospital transfer, a patient spends the acute-phase and recovery-phase hospitalization in a medical provider. In this case, the duration of hospital stay with the medical provider is $D^{in}$. However, with hospital transfer, the patient spends acute-phase and recovery-phase hospitalization with different medical providers. $D^{in}$ is the duration from the date of admission to the first medical provider to the date of discharge from the last medical provider. $H$ denotes a set of medical providers where a patient is hospitalized in a series of medical records.

**Network embedding.** The medical cooperation represented by the medical provider network contains various aspects of information contributing to the quality of healthcare provision. General network features, such as strength, represent only one aspect of the network. Thus, we use node2vec for feature engineering to represent the relationships among nodes in the structure of medical provider networks as features of medical cooperation.

node2vec is a method of embedding network structure into feature space to obtain feature representation of nodes using node sampling and skip-gram model [26]. Let $G = (V, E)$ denote the given network. node2vec samples nodes using second-order random walk controlled by two parameters, $p$ and $q$. When transitioning a random walker from the source node, the

transition probability from the $(i-1)$th node $v$ to $i$th node $x$:

$$P(x|v) \begin{cases} \dfrac{\pi_{vx}}{Z} & \text{if } (v,x) \in E \\ 0 & \text{otherwise} \end{cases}. \tag{4}$$

Here, $Z$ is the normalizing constant and $\pi_{vx}$ is the unnormalized transition probability. We introduced a search bias $\alpha_{pq}(u,x)$ to determine the probability $\pi_{vx}$ that a random walker transitions from node $v$ to node $x$ when it transitions from node $u$ to node $v$, as in the following equation:

$$\alpha_{pq}(u,x) = \begin{cases} \dfrac{1}{p} & \text{if } d_{ux} = 0 \\ 1 & \text{if } d_{ux} = 1 \\ \dfrac{1}{q} & \text{if } d_{ux} = 2 \end{cases}. \tag{5}$$

Here, $d_{ux}$ denotes the shortest path length between nodes $u$ and $x$. In this time, the transition probability $\pi_{vx} = \alpha_{pq}(u,x) \cdot w_{vx}$.

The parameters $p$ and $q$ controlling this second-order random walk are called the return parameter and the in-out parameter, respectively. Return parameter $p$ represents the likelihood that random walker transitions from $u$ to $v$ and then to $u$ again. In-out parameter $q$ controls whether the random walker performs inward or outward search. Node sampling by a random walker becomes a depth-first sampling (DFS)-like strategy or a breadth-first sampling (BFS)-like strategy controlling these parameters. Since BFS reflects homophily and DFS reflects structural equivalence of nodes in feature representation, we can obtain desirable feature representation by adjusting the $p$ and $q$ parameters to appropriate values. The length of the random walk $l$ and the number of walkers per node $t$ also need to be set. These parameters determine the number of samples for the following feature learning process.

We used a skip-gram model for feature learning. This method obtained a feature representation of each node by learning a model that predicts the neighboring nodes $N_s(u)$ of node $u$. Here, neighbor nodes refer to the nodes before and after node sequences obtained from the node sampling. The $w$ nodes before and after node $u$ were used as $N_s(u)$, where $w$ is the window size. Let one of the neighborhood nodes of node $u$ be $v \in N_s(u)$. Given the one-hot vector $\boldsymbol{x}_u(1 \times |V|$ vector) corresponding to the input node $u$, the probability of outputting one-hot vector $\boldsymbol{y}_v(|V| \times 1$ vector) corresponding to the output node $v$ is calculated using a softmax function, gives as

$$P(\boldsymbol{y}_v|\boldsymbol{x}_u) = \frac{\exp\phi(\boldsymbol{x}_u, \boldsymbol{y}_v)}{\sum_{v' \in V} \exp\phi(\boldsymbol{x}_u, \boldsymbol{y}_{v'})}. \tag{6}$$

Here, $\phi(\boldsymbol{x}_u, \boldsymbol{y}_v) = \boldsymbol{v}_u \cdot \boldsymbol{v}_v^T$ when the feature representations for nodes $u$ and $v$ are $\boldsymbol{v}_u$ and $\boldsymbol{v}_v(1 \times d$ vector), respectively. The dimension of the feature representation $d$ is given as a parameter. We assume conditional independence among the nodes in $N_s(u)$:

$$P(N_s(u)|\boldsymbol{x}_u) = \prod_{v \in N_s(u)} P(\boldsymbol{y}_v|\boldsymbol{x}_u). \tag{7}$$

When the input vector $\boldsymbol{x}_u$ is given for all nodes in $V$, we determine the feature representation $\boldsymbol{v}_u$ of node $u$ to maximize the likelihood of outputting the neighboring node $N_s(u)$. Feature

learning is the following log-likelihood maximization problem:

$$\max \sum_{u \in V} \log(P(N_s(u)|\boldsymbol{x}_u)). \tag{8}$$

Using Eqs (6) and (7), we represent Eq (8) as follows:

$$\max \sum_{u \in V} \left( -\log Z_u + \sum_{v \in N_s(u)} \boldsymbol{v}_u \cdot \boldsymbol{v}_v \right). \tag{9}$$

Here, $Z_u = \sum_{v \in V} \exp(\boldsymbol{v}_u \cdot \boldsymbol{v}_v)$. We obtained the feature representations $\boldsymbol{v}$ optimizing Eq (9) using the stochastic gradient ascent method. The hyperparameters in node2vec are $t, l, w, d, p, q$, and these need to be set.

**Regression model.** As shown in Fig 2(b), in case $i$, the patient is hospitalized in medical providers included in $H_i$. Let $\boldsymbol{c}_i$ denote the level of cooperation among the medical providers included in $H_i$. Examining the relationship between the level of medical cooperation and the duration of hospital stay $D_i^{\text{in}}$ at the case level indicates a regression problem. It is estimated using the following function $f$:

$$D_i^{\text{in}} = f(\boldsymbol{c}_i) + \varepsilon_i, \tag{10}$$

where $\varepsilon_i$ denotes the residual error. This model focuses only on the contribution of the medical cooperation factors $C$ and $Q = D^{\text{in}}$ in Eq (1). The predicted value $\hat{D}_i^{\text{in}} = f(\boldsymbol{c}_i)$ is calculated value by the function $f$ corresponding to the input variable $\boldsymbol{c}_i$.

When strength $s_j$ was used as a network feature to represent the level of medical cooperation of medical provider $j$, $\boldsymbol{c}_i$ is defined as the geometric mean of $s_j$ for medical providers included in $H_i$:

$$\boldsymbol{c}_i = \mu_{G_i}(s) = \prod_{j \in H_i} s_j^{1/n_i}. \tag{11}$$

Here, $n_i$ is the number of nodes in $H_i$, which represents the number of medical providers when the patient was admitted to the case $i$. In this case, $\boldsymbol{c}_i$ is one-dimensional input variable.

However, when we use feature representation $\boldsymbol{v}_j$, we define $\boldsymbol{c}_i$ as the mean of $\boldsymbol{v}_j$ of medical providers included in $H_i$:

$$\boldsymbol{c}_i = \mu_i(\boldsymbol{v}) = \frac{1}{n_i} \sum_{j \in H_i} \boldsymbol{v}_j \tag{12}$$

In this case, $\boldsymbol{c}_i$ is a $d-$dimensional input variable.

In the model with $\boldsymbol{c}$ defined by the feature representation $\boldsymbol{v}$ by node2vec as the input variables, we used a linear regression model and regression tree ensemble model. The regression tree ensemble model is a method for fitting a function $f$ using an ensemble of regression trees. The algorithms for the regression tree ensemble model are based on random forest [27, 28] and least-squares boosting (LSBoost) [29]. We use Bayesian optimization to optimize the method, which is random forest or LSBoost, and hyperparameters. [30–32].

Given $n$ pairs of output variable $y$ and input variables $\boldsymbol{x}$, the regression tree $T$ fits $y$ with the terminal node $R_i$ ($i = 1, 2, \ldots, I$), which is the node at the end of tree splitting, as follows:

$$T(\boldsymbol{x}) = \sum_{i=1}^{I} \gamma_i I(\boldsymbol{x} \in R_i). \tag{13}$$

Here, $\gamma_i$ is a constant determined for each terminal node $R_i$. We repeated the following process to obtain $T$ until the minimum number of samples at the terminal node $n_{\min}$ or the maximum number of splits $s_{\max}$ is reached:

- Randomly select $i$ from the input variables.

- Choose the best variable and split point among $i$.

- Split the node into two sub-nodes.

Ensemble learning is a statistical model that uses an ensemble as a learner of trees $T$ obtained by the above-mentioned process. Here, let the output variable $y = D^{\text{in}}$ and the input variables $\boldsymbol{x} = \boldsymbol{c}$. We obtained $M$ bootstrap replicas $Z^*$ by randomly selecting $n$ samples extracted from $n$ samples. Tree $T_m$ is trained for each $Z_m^*$. The random forest algorithm uses the average of $M$ trees $T_m$ as the ensemble learner:

$$\hat{D}^{\text{in}} = \frac{1}{M} \sum_{m=1}^{M} T_m(\boldsymbol{c}). \tag{14}$$

LSBoost considers the loss function $L$ as the mean square error and aggregates the new learner to all the previously trained learners in order to minimize the loss function $L$ at each step:

$$L(D_i^{\text{in}}, f(\boldsymbol{c}_i)) = \frac{1}{2}(D_i^{\text{in}} - f(\boldsymbol{c}_i))^2. \tag{15}$$

The detailed procedure is as follows: First, we initialized $f_0(\boldsymbol{c})$ to

$$f_0(\boldsymbol{c}) = \arg \min_{\gamma} \sum_{i=1}^{n} L(D_i^{\text{in}}, \gamma). \tag{16}$$

This represents a constant model with only one leaf. We performed the following steps from $m = 1$ to $M$.

$$r_{im} = -\left[\frac{\partial L(D_i^{\text{in}}, f(\boldsymbol{c}_i))}{\partial f(\boldsymbol{c}_i)}\right]_{f=f_{m-1}} = D_i^{\text{in}} - f_{m-1}(\boldsymbol{c}_i) \ (i = 1, 2, \ldots, n). \tag{17}$$

We fitted the regression tree $T_m$ to the $r_{im}$. We calculated the following equation using terminal nodes $R_{jm}(j = 1, 2, \ldots, J_m)$ of $T_m$:

$$\gamma_{jm} = \arg \min_{\gamma} \sum_{\boldsymbol{c} \in R_{jm}} L(D_i^{\text{in}}, f_{m-1}(\boldsymbol{c}_i) + \gamma) \ (j = 1, 2, \ldots, J_m). \tag{18}$$

Using this $\gamma_{jm}$, we update $f_m$ as shown in the equation below:

$$f_m(\boldsymbol{c}) = f_{m-1}(\boldsymbol{c}) + \eta \sum_{j=1}^{J_m} \gamma_{jm} I(x \in R_{jm}) \tag{19}$$

Here, $\eta$ is a parameter that controls the learning rate for each step, and $0 < \eta \leq 1$. We obtain the regression function $f_M(\boldsymbol{c})$ by repeating this step $M$ times. Using this function, we output $\hat{D}_{in} = f_M(\boldsymbol{c})$.

**Considered models.** We investigated 4 models using these methods:

**Model 1** Output variable: duration of hospital stay $D^{\text{in}}$, input variable: geometric mean of strength $\mu_G(s)$, regression model: linear regression.

**Model 2** Output variable: duration of hospital stay $D^{in}$, input variable: geometric mean of strength $\mu_G(s)$, age at time of admission and regression model: linear regression.

**Model 3** Output variable: duration of hospital stay $D^{in}$, input variable: mean of feature representation using node2vec $\mu(\boldsymbol{v})$, regression model: linear regression.

**Model 4** Output variable: duration of hospital stay $D^{in}$, input variable: mean of feature representation using node2vec $\mu(\boldsymbol{v})$, regression model: regression tree ensemble.

Model 1 explains the duration of hospital stay, $D^{in}$, using a geometric mean of strength $\mu_G(s)$ representing the explicit level of medical cooperation. Model 2 checks whether there is a contribution of medical cooperation when the age at admission was incorporated as an input variable in Model 1. Model 3 and Model 4 use mean of the feature representation using node2-vec $\mu(\boldsymbol{v})$. $\boldsymbol{v}$ represents the mapping of the relationship between each medical provider on the network representing the cooperation among medical providers in the feature space. When the input variables were extended to the $d$–dimension, we expected a higher explanatory ability for the contribution of medical cooperation to the quality of healthcare provision than the one-dimensional $s$. Furthermore, Model 4 uses ensemble learning as a statistical model to examine the extent to the medical cooperation factor can explain the variation in the duration of hospital stay $D^{in}$.

## Community analysis

We used community analysis to divide the network into densely connected sub-networks and further investigate regional medical inter-institutional cooperation in the prefecture. We used Infomap for community analysis. Infomap is a method used to detect communities by optimizing the map equation as an evaluation function [33–35]. Infomap efficiently encodes the trajectories of random walkers in a network. We use the fact that a random walker stays in a community for a long time under the ideal code. The ideal coding procedure is as follows:

- Assign one code to each community.

- Assign one code to each node within each community.

- Assign exit code when the walker leaves a community.

The community structure is obtained from the code assigned to each community. The optimal code can be obtained by minimizing the map equation:

$$\mathcal{L} = qH(\mathcal{Q}) + \sum_{c=1}^{n_c} p_{\circlearrowleft}^c H(P_c), \tag{20}$$

where $n_c$ denotes the number of communities. The first term of Eq (20) corresponds to the average number of bits to describe the movement between communities. The second term of Eq (20) represents the average number of bits used to describe the movement within a community.

$\mathcal{L}$ takes a specific value depending on the partitioned community structure. We must minimize $\mathcal{L}$ for all partitions to determine the optimal partition. Louvain's algorithm [36] was used for efficient optimization. We assigned each node to a separate community. If $\mathcal{L}$ decreases when neighboring nodes are joined, we join the nodes. By repeating this process, we can efficiently perform community detection by Infomap.

## Results

Firstly, we present the basic features of the constructed network. We also describe the linear regression model analysis results with strength as the input variable, linear regression model, and regression tree ensemble model with the feature representation using node2vec as the input variable. Finally, we examine the relationship between medical cooperation and networks structure using community analysis.

### Constructed network

We analyzed the maximum connected component of the medical provider network. The prefecture is divided into five medical administration areas, each of which is designed to provide general inpatient care. The network diagram Fig 3 shows that the medical providers in each medical administration areas tend to cooperate and were clustered close to the network. Note that we classified out-of-prefecture medical providers used by patients in the prefecture as others. In addition, the nodes in Fig 3 were sized according to the number of beds, indicating that the larger medical providers in the prefecture occupy the central positions in the network.

Table 2 lists the network features, and Fig 4(a) and 4(b) show the cumulative distributions of degree and strength. The network features reveal the structural characteristics of the medical provider network. The average clustering coefficient is 0.535. This indicates that the medical providers tended to form clusters with each other. The assortativity was negative. This implies that the network is disassortative. These indicate that the central hospital in an area serves as a hub for medical cooperation and shares patients with smaller medical providers in the periphery of the network.

### Linear regression model using strength

We examined the relationship between medical cooperation and the duration of hospital stay $D^{in}$ using a linear regression model. The strength of each medical provider was the value representing the level of medical cooperation (Model 1 and 2). We included cases of hospitalization with surgery at a medical provider for the series of medical records. In addition, among the target cases, hospitalized patients are in the maximum-connected components of the network. Some elderly patients do not undergo surgery even if their fracture requires hospitalization due to their physical condition. Therefore, cases without surgery were excluded. Additionally, cases in which the duration of hospital stay $D^{in}$ exceeded 400 days were excluded from the study. Table 3 presents information on the target cases. The total number of cases was 2,332. The average age of patients at admission was 83.2 years, and approximately 80% of the patients were female. In 80% of cases, patients were admitted to the same medical provider for the acute and recovery phases without transfer. In about 20% of cases, patients were transferred between two or more medical providers.

The average value of the duration of hospital stay $D^{in}$ was 65.6 days. Fig 5(a) and 5(b) show the distribution of $D^{in}$. Both tails are similar to power law distribution. We calculated the $\mu_G(s)$ of medical providers hospitalized by a patient in each case from the strength $s$ of individual medical providers using Eq (11). We used $\mu_G(s)$ as an input variable. The distribution of the strength $s$ is shown in Fig 4(b). In addition to $\mu_G(s)$, we also considered a model with age as an input variable. This is because age is related to the damage caused by fractures and the time required for recovering.

Tables 4 and 5 show the results of the regression analysis for Model 1 and 2. In both analyses, the p-value of the t-test for the coefficient of each $\mu_G(s)$ was significantly negative at the 1% level. We found that patients admitted to medical providers with a higher average level of cooperation among medical providers had a shorter duration of hospital stay $D^{in}$. The same

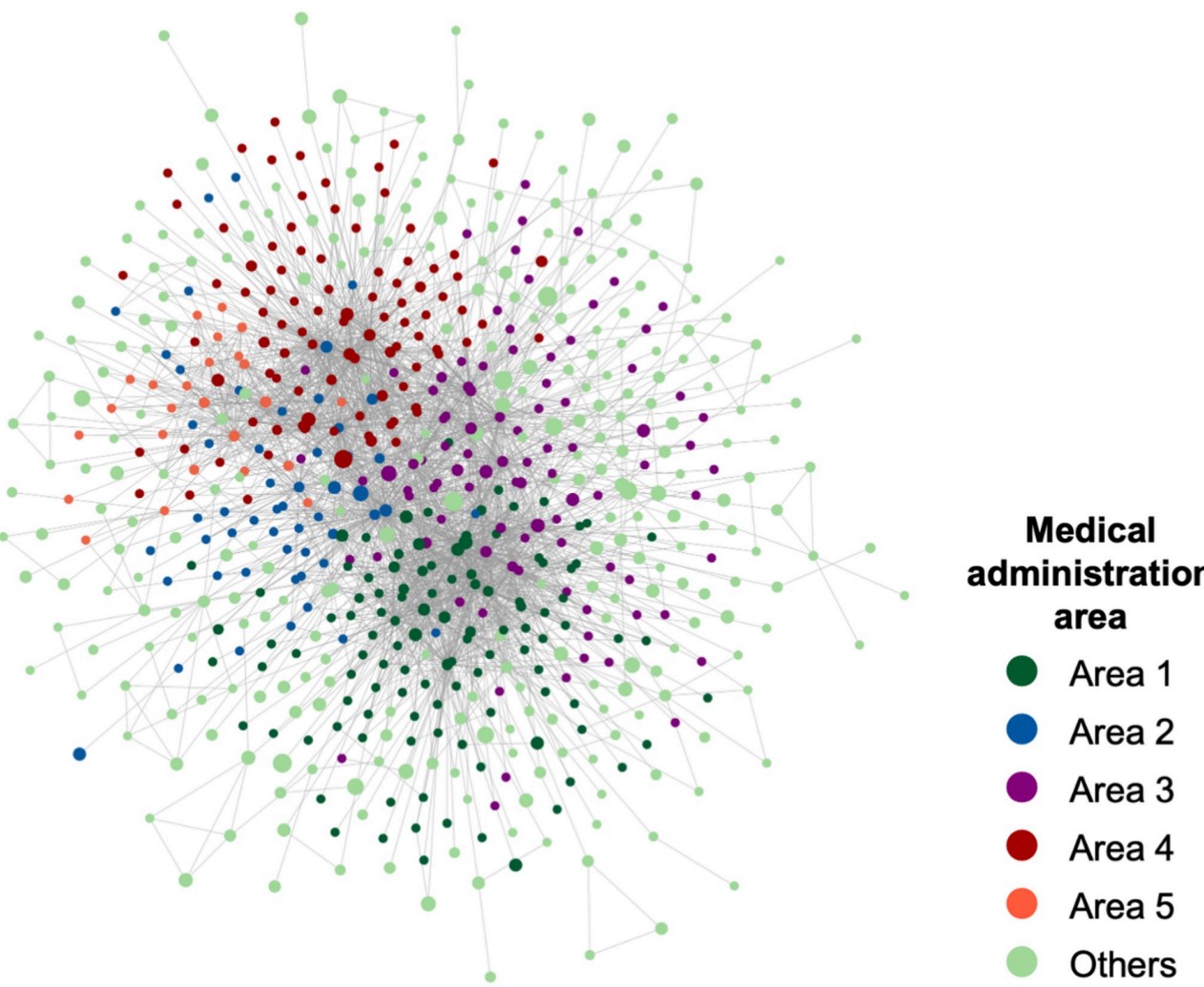

**Fig 3. Network diagram of the medical provider network.** Number of nodes (medical providers) $N = 644$, number of edges $L = 2756$. The prefecture is divided into five medical administration areas, and the nodes are color-coded according to each area. The node size is based on the number of beds in the medical provider. The nodes located in the same area are close to each other on the network.

**Table 2. Network features of medical providers network.**

| Network feature | Calculated value |
|---|---|
| Number of nodes | 644 |
| Number of edges | 2756 |
| Average degree | 8.3 |
| Average strength | 1.3 |
| Average distance | 3.09 |
| Average clustering coefficient | 0.535 |
| Assortativity | -0.258 |

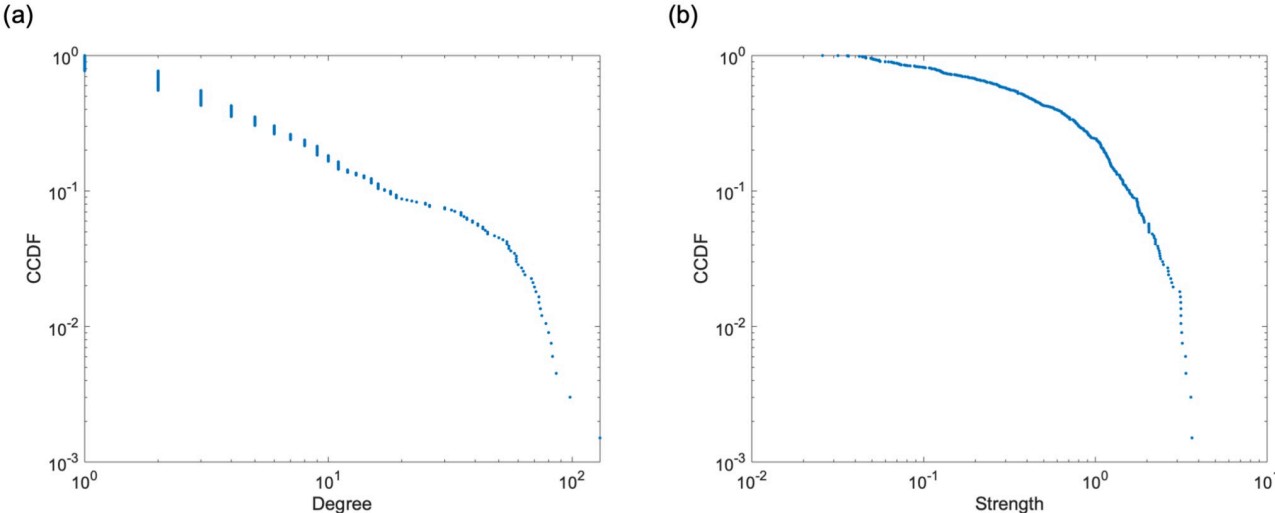

**Fig 4. Empirical complementary cumulative distribution functions (CCDF) of degree and strength.** (a) Degree and (b) strength. Both are widely distributed. These figures show that the number of medical providers with which patients are shared and the level of medical cooperation with neighboring medical providers vary greatly among medical providers.

relationships can be observed in Fig 6(a) and 6(b). The explanatory ability of the statistical model improved when age was added to the input variables to account for the effect of increasing the duration of hospital stay $D^{in}$ due to aging ($R^2$ = 0.0085 and 0.011, respectively). In contrast, the coefficients of determination $R^2$ are low in both models, and the statistical model did not have a high explanatory ability for the duration of hospital stay $D^{in}$. Fig 7 suggests that Model 1 and 2 are underfitting, indicating that the statistical model lacks expressive abilities. We also compared the results with the arithmetic mean of the strength $\mu(s)$, and confirmed that $\mu_G(s)$ has the better explanatory ability (S2 Fig).

## Linear regression model using feature representation

We performed regression using a statistical model with the feature representation $v$ of each medical provider using node2vec as the input variable. The output variable was $D^{in}$, and the input variable was the mean value of $v$ among the medical providers included in $H_i$, as shown in Eq (12).

We determined the hyperparameters of node2vec in the following way: As pre-analysis, we set the default parameter values as $(t, l, w, d, p, q)$ = (5, 25, 10, 70, 1, 1). We extracted feature

**Table 3. Information of targeted cases.**

| Items | | Value |
|---|---|---|
| Number of cases | | 2332 |
| Age | Mean | 83.2 |
| Duration of hospital stay $D^{in}$ | Mean | 65.6 |
| Sex | Male | 495(21.2%) |
| | Female | 1837(78.8%) |
| Number of hospitalized medical providers | 1 | 1898(81.4%) |
| | 2 | 401(17.2%) |
| | >3 | 33(1.4%) |

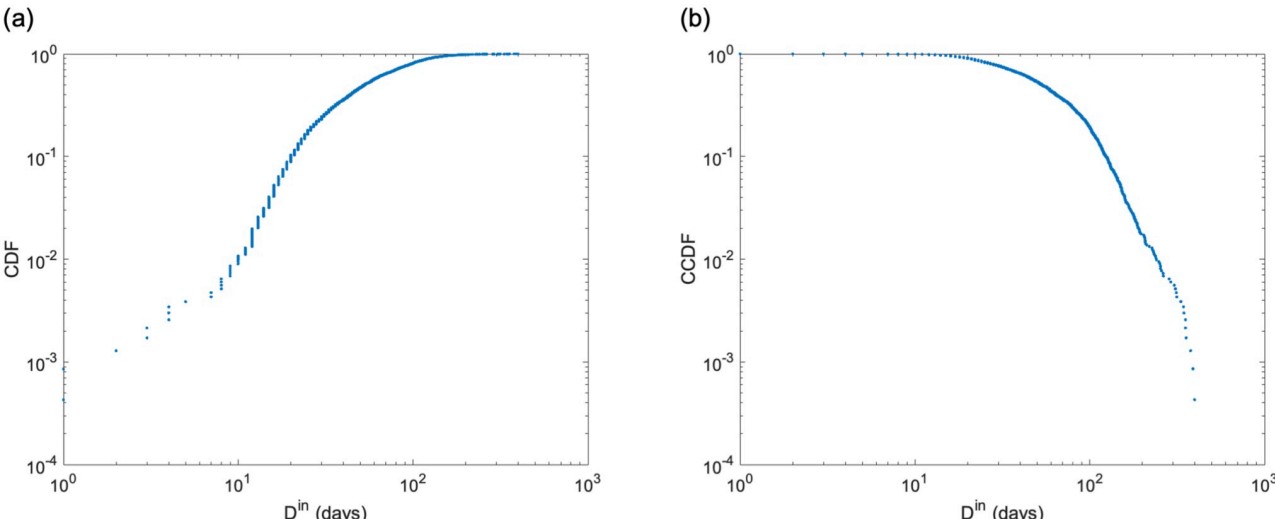

**Fig 5. Distribution of duration of hospital stay $D^{in}$.** (a) Empirical cumulative distribution functions (CDF). (b) Empirical complementary cumulative distribution functions (CCDF). In the logarithmic plot, both tails are straight, which are similar to a power law distribution.

representations by adjusting $t$, $l$, $w$, and $d$ using one parameter each. Regression analysis was performed using the extracted $\boldsymbol{v}$. We calculated root mean squared error (RMSE) using $\hat{D}^{in}$ obtained from the regression analysis.

$$\text{RMSE} = \sqrt{\left(D^{in} - \hat{D}^{in}\right)^2} \tag{21}$$

RMSE represents the fitting performance of the fitting of the regression model. In the pre-analysis, we performed 20 iterations with five-fold cross-validation and used the average value of RMSE as the evaluation value of the regression performance. As a result, we used the parameter $(t, l, w, d) = (7, 60, 13, 75)$, where the mean value of RMSE is the minimum value of each parameter. The results for this pre-analysis are shown in S3 Fig.

After the pre-analysis, we performed a grid search for parameters $p$ and $q$. The range of $p$ and $q$ explored are $p, q \in \{1/8, 1/4, 1/2, 1, 2, 4, 8\}$, respectively. The evaluation is the RMSE of the regression analysis with five-fold cross-validation. The average value of the RMSE over 100 iterations is shown in Fig 8(a). Optimal combination of the parameters were $(p, q) = (8, 0.25)$. From the result of the grid search, we determined that the regression performance tends to improve when $p$ is large and $q$ is small.

We expanded the search range to $p \in \{4, 8, 16, 32, 64, 128\}$ and $q \in \{1/128, 1/64, 1/32, 1/16, 1/8, 1/4\}$ including the optimal parameters of the first grid search and performed a second grid

**Table 4. Regression analysis of duration of hospital stay $D^{in}$ by geometric mean of strength $\mu_G(s)$.**

|  | Coefficient | Std. error | t | p-value (t) |
|---|---|---|---|---|
| (Intercept) | 78.02 | 2.96 | 26.36 | <0.01 |
| Geometric mean of strength $\mu_G(s)$ | -5.72 | 1.28 | -4.46 | <0.01 |
| $R^2$ | 0.0085 | F | 19.89 |  |
| $R^2$ (5-fold CV)* | 0.0075 | p-value (F) | <0.01 |  |
| Adjusted $R^2$ | 0.0080 |  |  |  |

*$R^2$ calculated from $\hat{D}^{in}$ of the test data with five-fold cross-validation. We calculated it for comparison with models using ensemble learning.

**Table 5. Regression analysis of duration of hospital stay $D^{in}$ by geometric mean of strength $\mu_G(s)$ and age at admission.**

| | Coefficient | Std. error | t | p-value (t) |
|---|---|---|---|---|
| (Intercept) | 56.48 | 8.87 | 6.37 | <0.01 |
| Geometric mean of strength $\mu_G(s)$ | -5.67 | 1.30 | -4.43 | <0.01 |
| Age | 0.26 | 0.10 | 2.58 | 0.01 |
| $R^2$ | 0.011 | F | 13.28 | |
| $R^2$ (5-fold CV)* | 0.0097 | p-value (F) | <0.01 | |
| Adjusted $R^2$ | 0.010 | | | |

*$R^2$ calculated from $\hat{D}^{in}$ of the test data with five-fold cross-validation. We calculated it for comparison with models using ensemble learning.

search. In addition, we increased the number of iterations to 300 to smoothen the evaluation values. Fig 8(b) shows the results. We obtained $(p, q) = (16, 1/32)$ as the optimal parameters, whereas there is little change in the RMSE where $p$ is sufficiently large and $q$ is sufficiently small. As shown in the surface plot in S4 Fig, the RMSE becomes planar as $p$ increases and $q$ decreases. It does not take a unique optimum value and has many local minimum points.

The $p$ and $q$ parameter controls the probability of selecting the next transition node $x$ when transitioning from node $u$ to node $v$, as shown in Eq (5). When $p$ is large, and $q$ is small, the random walker tends not to return to node $u$ and transits to node $x$ with $d_{ux} = 2$. The setting was a DFS-like sampling strategy. These results indicate that the regression performance is improved when the parameter settings are closer to DFS. However, after setting the parameters sufficiently close to DFS, the performance did not improve and got independent of parameters $p$ and $q$.

The regression analysis of $D^{in}$ with the optimal parameters $(t, l, w, d, p, q) = (7, 60, 13, 75, 16, 1/32)$ obtained from the grid search showed that the mean value of the coefficient of determination $R^2$ was 0.10, and the highest value was 0.17. Fig 9 shows a plot of the predicted value $\hat{D}^{in}$ and measured values $D^{in}$ when the $R^2$ had the highest value. Compared with the model

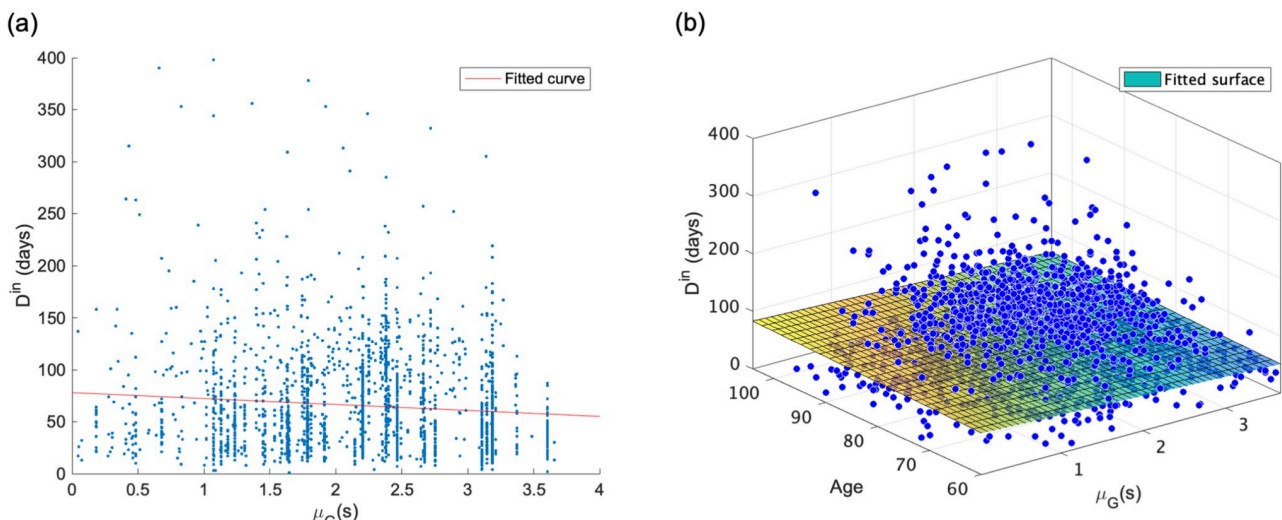

**Fig 6. Result of linear regression model with duration of hospital stay $D^{in}$ as output variable and geometric mean of strength $\mu_G(s)$ and age as input variables.** (a) $D^{in}$ vs. $\mu_G(s)$ ($R^2 = 0.0085$). (b) $D^{in}$ vs. $\mu_G(s)$ and age ($R^2 = 0.011$). These show the negative relationship between $D^{in}$ and $\mu_G(s)$. In addition, the explanatory ability of the statistical model is increased by considering the effect of age on the duration of of hospital stay.

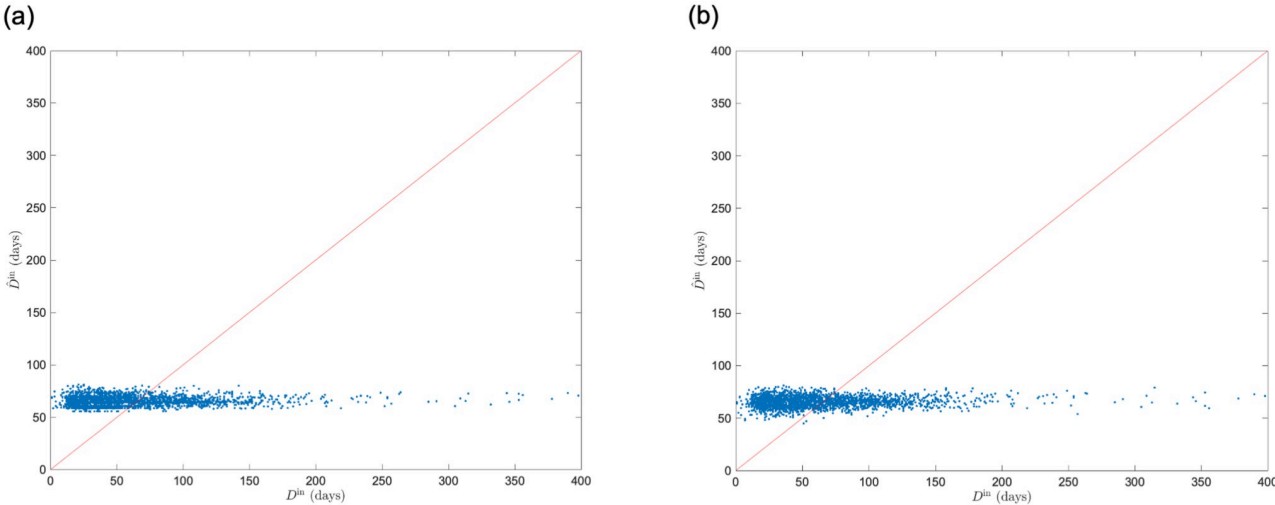

**Fig 7. Comparison of measured duration of hospital stay $D^{in}$ and predicted the duration of of hospital stay $\hat{D}^{in}$ by linear regression model with geometric mean of strength $\mu_G(s)$ and age as input variables.** (a) $\mu_G(s)$. (b) $\mu_G(s)$ and age. $\hat{D}^{in}$ obtained by five-fold cross validation is shown. Both are underfitting and do not fully explain the variation in $D^{in}$.

with $\mu_G(s)$ as the input variable, the performance was improved approximately ten-fold. We found that the causes of decreasing regression performance were variation in $\hat{D}^{in}$ around $D^{in} = 0$ and underestimation in domains with large $D^{in}$.

## Regression tree ensemble model using feature representation

We performed an analysis using the regression tree ensemble model with the same $\nu$ used in the linear regression. The algorithms and hyperparameters of regression tree ensemble model were optimized using Bayesian optimization. The mean value of the coefficient of

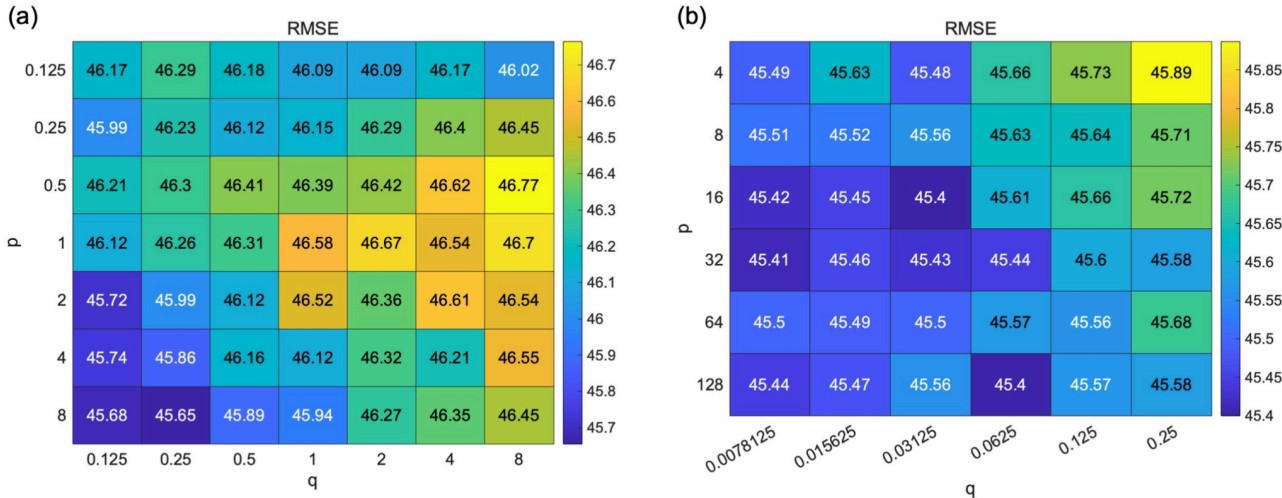

**Fig 8. Result of grid search of hyperparameters $p$ and $q$ of node2vec.** (a) $p, q \in \{1/8, 1/4, 1/2, 1, 2, 4, 8\}$, the number of iteration = 100. (b) $p \in \{4, 8, 16, 32, 64, 128\}$ and $q \in \{1/128, 1/64, 1/32, 1/16, 1/8, 1/4\}$, the number of iteration = 300. Here, we set $(t, l, w, d) = (7, 60, 13, 75)$, as determined by the pre-analysis. We evaluated the regression performance with the changes in parameters $p$ and $q$ using the root mean squared error (RMSE). Although the variation in RMSE with the change in parameters is not large, the regression performance tends to improve when $p$ is large, and $q$ is small. However, there is little change in RMSE in the range when $p$ is sufficiently large and $q$ is sufficiently small.

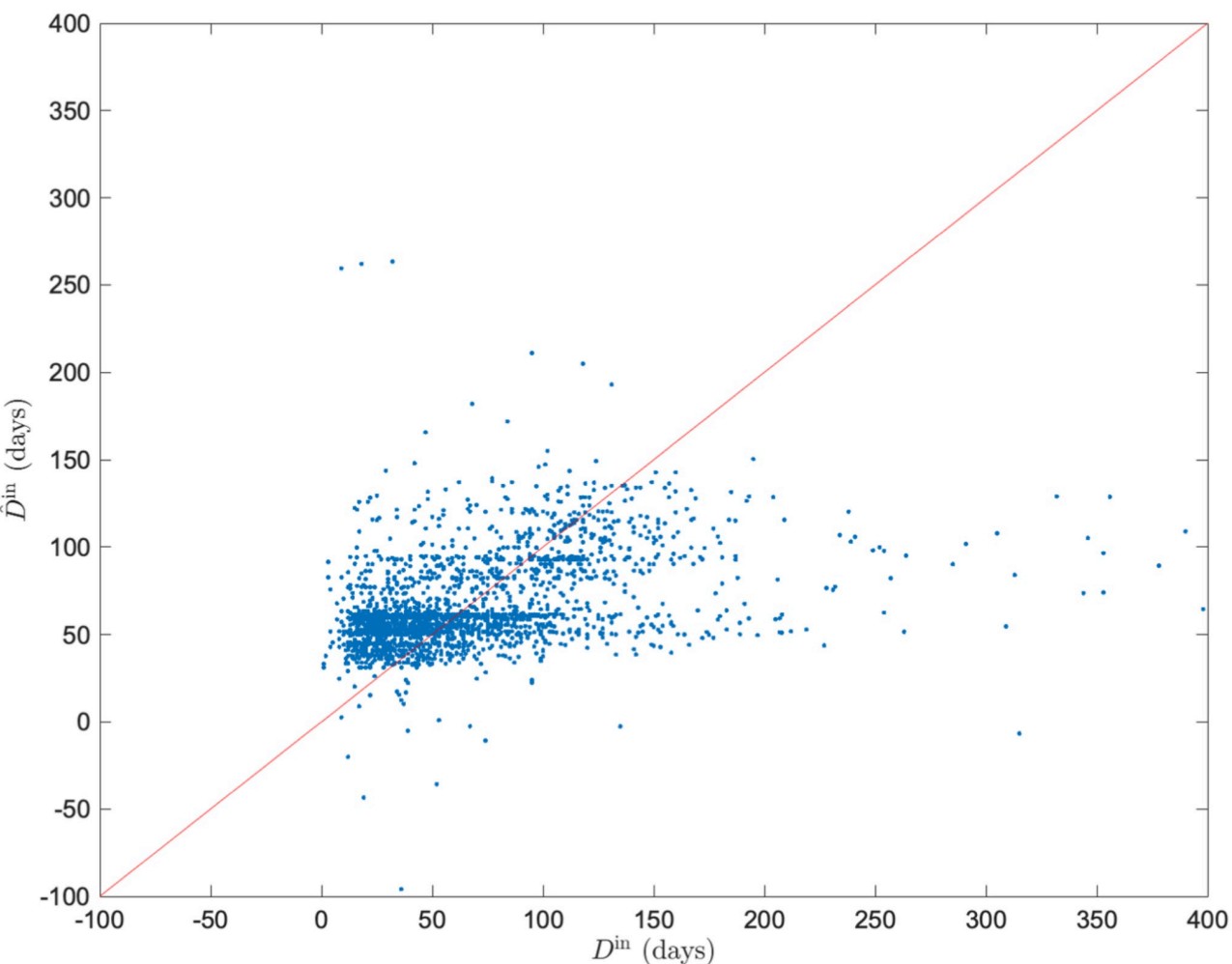

**Fig 9. Comparison of measured duration of hospital stay $D^{\text{in}}$ and predicted duration of hospital stay $\hat{D}^{\text{in}}$ calculated by linear regression model.**
The predicted and the measured values calculated by the linear regression model with the average value of $\mu(v)$ of the feature representations by node2vec as the input variable. We select the best result comparing the coefficient of determination $R^2$ among all sets extracted through 300 iterations ($R^2 = 0.17$).

determination $R^2$ was 0.21, and the highest value was 0.24. Fig 9 shows a plot of the predicted value $\hat{D}^{\text{in}}$ and the measured values $D^{\text{in}}$ when the $R^2$ had the highest value. We found that the performance improved by approximately two-fold compared to the linear regression model, which is Model 3. The performance was approximately 20 times better than that of the model using $\mu_G(s)$, which is Model 1. The tree ensemble model improves the performance of $\hat{D}^{\text{in}}$ around $D^{\text{in}} = 0$ based on a comparison of Figs 8 and 9. However, there is still a tendency to underestimate $\hat{D}^{\text{in}}$ in domains with large $D^{\text{in}}$.

## Medical cooperation and network structure

Based on the results of the regression analysis, we found a relationship between the geometric mean of strength $\mu_G(s)$ among medical providers and the duration of hospital stay $D^{\text{in}}$. We considered that strength $s$ is adopted as a value representing the level of cooperation among

medical providers from this relationship. To examine the average level of medical cooperation in each community, we identified communities and calculated each community's $\mu_G(s)$.

We used a two-level Infomap and detected 40 communities. Since 95% of the medical providers in the prefecture were in the seven communities with the largest number of components, we compared the $\mu_G(s)$ among these seven communities. We also compared $\mu_G(s)$ and other network features of each community. Table 6 shows the results. We calculated $\mu_G(s)$ for all medical providers belonging to a community and only those in the prefecture. We found a disparity in medical cooperation among communities, which was more pronounced when we focused only on medical providers in the prefecture.

We explain the cause of this disparity from the relationship between the network features and $\mu_G(s)$. Fig 10 shows there is a linear relationship between network features and $\mu_G(s)$. Fig 10(a) shows the relationship between $\mu_G(s)$ and the average distance normalized by the

**Table 6. Network features of communities.**

| | | Community 1 | Community 2 | Community 3 | Community 4 |
|---|---|---|---|---|---|
| Number of nodes | Total | 196 | 152 | 64 | 45 |
| | In-prefecture | 135(68.9%) | 108(71.1%) | 40(62.5%) | 35(77.8%) |
| | Area 1 | 99(50.5%) | 0(0.0%) | 0(0.0%) | 5(11.1%) |
| | Area 2 | 1(0.5%) | 5(3.3%) | 0(0.0%) | 27(60.0%) |
| | Area 3 | 35(17.9%) | 3(2.0%) | 32(50.0%) | 1(2.2%) |
| | Area 4 | 0(0.0%) | 77(50.7%) | 8(12.5%) | 2(4.4%) |
| | Area 5 | 0(0.0%) | 23(15.1%) | 0(0.0%) | 0(0.0%) |
| | Out-of-prefecture | 61(31.1%) | 44(28.9%) | 24(37.5%) | 10(22.2%) |
| Number of edges | | 848 | 606 | 143 | 89 |
| Average degree | | 8.65 | 7.97 | 4.47 | 3.96 |
| Average strength | | 0.99 | 1.01 | 0.79 | 0.54 |
| Average distance | | 2.53 | 2.43 | 2.28 | 2.07 |
| Average clustering coefficient | | 0.51 | 0.63 | 0.62 | 0.57 |
| Assortativity | | -0.40 | -0.40 | -0.57 | -0.61 |
| Geometric mean of strength | | 0.30 | 0.31 | 0.30 | 0.18 |
| | In-prefecture | 0.39 | 0.38 | 0.34 | 0.23 |
| | | Community 5 | Community 6 | Community 7 | |
| Number of nodes | Total | 31 | 18 | 12 | |
| | In-prefecture | 20(64.5%) | 12(66.7%) | 8(66.7%) | |
| | Area 1 | 1(3.2%) | 0(0.0%) | 0(0.0%) | |
| | Area 2 | 0(0.0%) | 7(38.9%) | 8(66.7%) | |
| | Area 3 | 18(58.1%) | 0(0.0%) | 0(0.0%) | |
| | Area 4 | 1(3.2%) | 5(27.8%) | 0(0.0%) | |
| | Area 5 | 0(0.0%) | 0(0.0%) | 0(0.0%) | |
| | Out-of-prefecture | 11(33.3%) | 6(33.3%) | 4(33.3%) | |
| Number of edges | | 42 | 21 | 14 | |
| Average degree | | 2.71 | 2.33 | 2.33 | |
| Average strength | | 0.50 | 0.95 | 0.50 | |
| Average distance | | 2.12 | 1.95 | 1.89 | |
| Average clustering coefficient | | 0.39 | 0.24 | 0.37 | |
| Assortativity | | -0.74 | -0.71 | -0.69 | |
| Geometric mean of strength | | 0.18 | 0.22 | 0.21 | |
| | In-prefecture | 0.20 | 0.17 | 0.28 | |

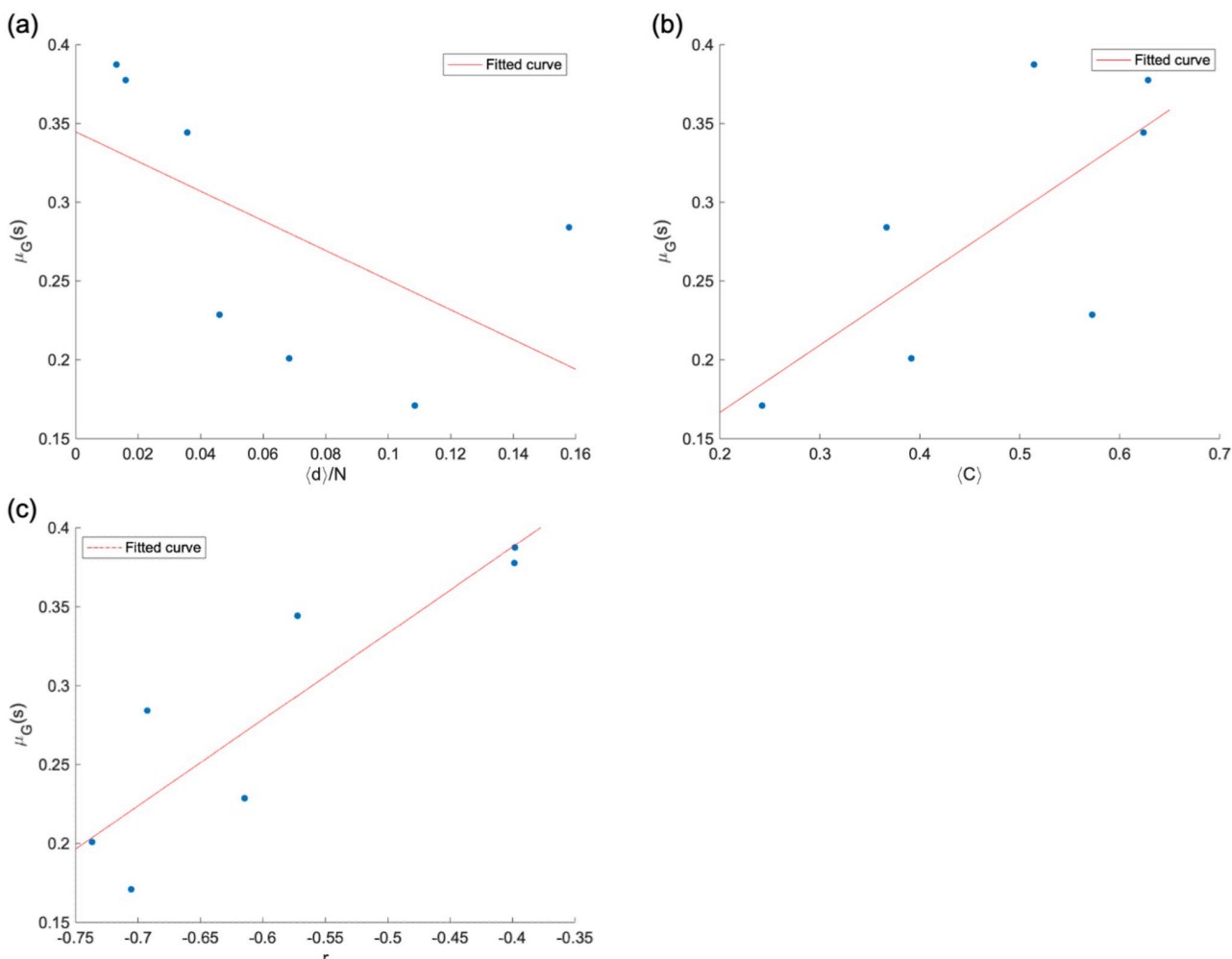

**Fig 10. Relationship between geometric mean of strength $\mu_G(s)$ and network features of each community.** (a) $\mu_G(s)$ vs. the average distance normalized by the number of nodes $\langle d \rangle/N$. (b) $\mu_G(s)$ vs. the average clustering coefficient $\langle C \rangle$. (c) $\mu_G(s)$ vs. assortativity $r$. There is a linear relationship between each network feature and $\mu_G(s)$, indicating that there is a relationship between network structure and medical cooperation.

number of nodes, $\langle d \rangle/N$. The closer the medical providers in the network, the higher is the level of medical cooperation. Fig 10(b) shows the relationship between $\mu_G(s)$ and the average clustering coefficient $\langle C \rangle$, where communities with a higher clustering tendency have a higher level of medical cooperation. Fig 10(c) shows the relationship between $\mu_G(s)$ and assortativity $r$, where communities with less disassortativity have a stronger level of medical cooperation. This indicates that a horizontally decentralized structure where each medical provider is connected to the other is more suitable for medical cooperation than a centralized structure in which the edge is concentrated around a particular medical provider.

Each community comprises medical providers located in a specific area, indicating the mutual collaboration among medical providers in close geographical distance. In addition, the results of the medical providers included in each community by medical administration areas are listed in Table 6. This result indicates that the community structure reflects the geographical distance. It confirms the observation from the network diagram shown in Fig 3, in terms of community analysis.

## Discussion

This study aimed to examine the overall contribution to the quality of health care provision from medical inter-institutional cooperation. For this purpose, we considered the four types of statistical models. Table 7 presents a comparison of the results of each regression model. In the table, we compare the $R^2$ and adjusted $R^2$ values of the test data from five-fold cross-validation for each model considering the differences in regression methods and dimensions of input variables.

Based on Model 1, we found a negative relationship between the geometric mean of the strength and the duration of hospital stay in medical providers. Patients with femoral neck fractures were admitted into the hospital. In addition, Model 2 shows that medical cooperation is effective even when the patient's age at admission as a factor affecting prolonged hospital stay was considered. These results indicate that the general network features contribute to the healthcare quality at the case level, which have been investigated in previous studies on medical provider networks at the patient level [13, 14]. Contrarily, the explanatory ability of statistical models for Models 1 and 2 was weak ($R^2$ = 0.0085, 0.011, respectively), and Fig 7 confirms that underfitting is the reason for the low explanatory ability of the statistical models.

Strength represents the summation of the cosine similarities of patients shared with neighboring medical providers in the network. The cooperation with neighboring medical providers to adjust capacity to appropriately care for patients may function effectively and shorten the duration of hospital stay. However, it occurs only if a patient hospitalizes in medical providers that frequently share patients with neighboring medical providers. The time from the fracture to surgery was identified as a risk factor for femoral neck fractures [20, 22] and capacity adjustment in the acute phase hospitals is essential. This implies that the results of the study examining impact of network structure on A&E performance in the analysis of patient transfer networks between wards may also be appropriate for regional medical cooperation [37].

We also examined Model 3 as a model with enhanced ability to represent input variables using feature representation by node2vec. We determined the parameters $p$ and $q$ of node2vec using a grid search. Fig 8 shows that the performance of the regression model is improved when $p$ is large and $q$ is small. It means that the probability of returning is low, and exploring outward is high when a random walker samples nodes, and the sampling strategy was DFS-like. DFS reflects a structural equivalence of the nodes in the embedding into the feature space, whereas the BFS reflects homophily [26]. We found that information on structural similarity (rather than the relationship with neighboring nodes closely interacting in the network structure) represents medical cooperation in medical provider networks. This result suggests that nodes structurally similar in a medical provider network play an important role in maintaining healthcare quality. The results of Model 3 show that using node2vec feature representations as input improved performance approximately ten times compared to Model 2 (mean value of

**Table 7. Comparison of regression performance among four models.**

| | | $R^2$ (5-fold CV) | Adjusted $R^2$ (5-fold CV) |
|---|---|---|---|
| Model 1 | | 0.0075 | 0.0071 |
| Model 2 | | 0.010 | 0.0088 |
| Model 3 | Mean | 0.10 | 0.066 |
| | Best | 0.17 | 0.15 |
| Model 4 | Mean | 0.21 | 0.18 |
| | Best | 0.24 | 0.21 |

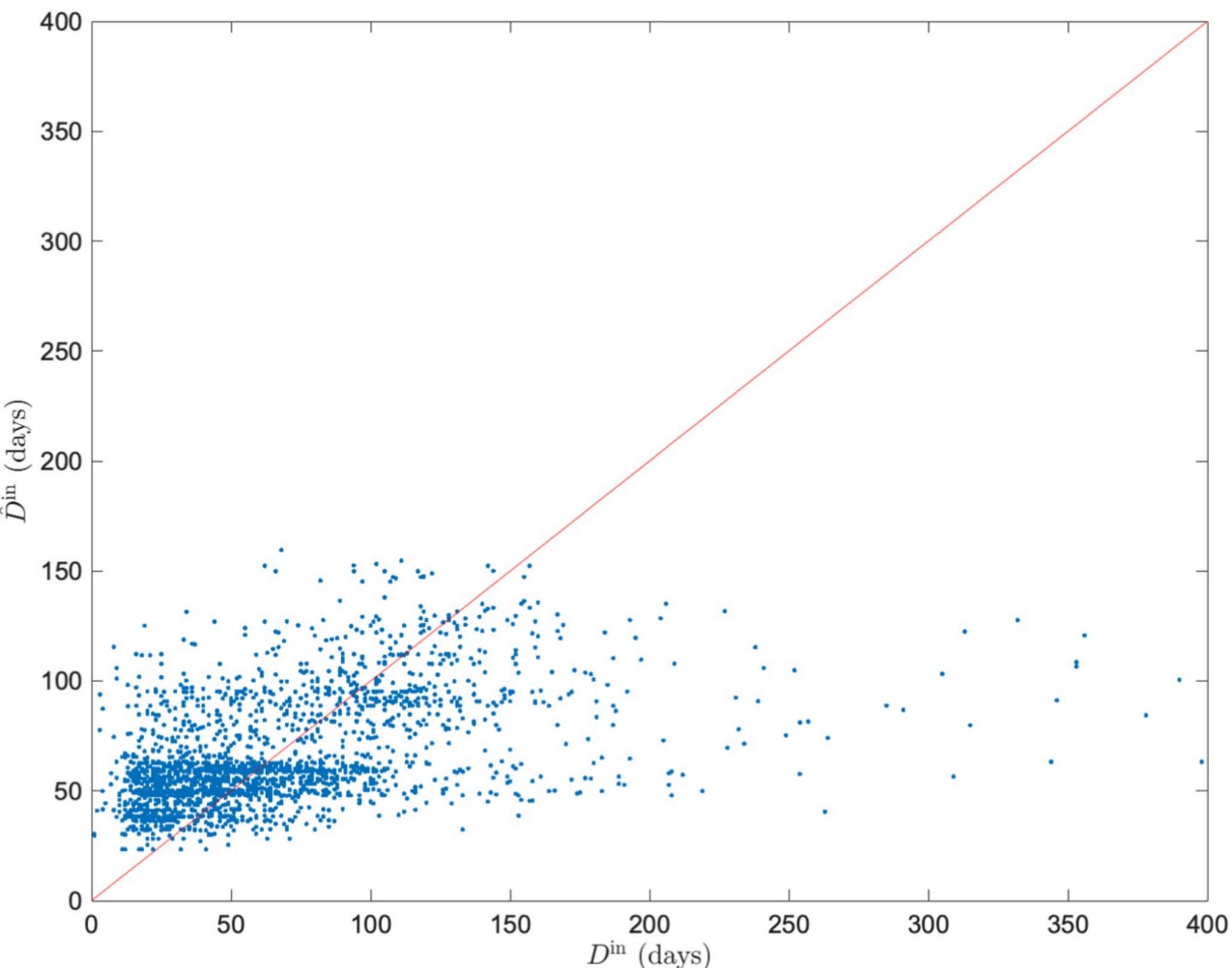

**Fig 11. Comparison of measured duration of hospital stay $D^{in}$ and predicted duration of hospital stay $\hat{D}^{in}$ calculated by regression tree ensemble model.** The predicted and measured values calculated by the regression tree ensemble model with the average value of $\mu(v)$ of the feature representations by node2vec as the input variable. We select the best result comparing the coefficient of determination $R^2$ among all sets extracted through 300 iterations ($R^2 = 0.24$).

$R^2 = 0.10$). Thus, the strength does not fully represent the information about medical cooperation contained in the medical provider network. node2vec fully extracted information as a feature. However, when $D^{in}$ was approximately 0, the value of $\hat{D}^{in}$ varied greatly from Fig 9. This is a factor of performance loss.

We used the regression tree ensemble model to improve the regression performance in Model 4. We improved the performance by approximately two times that of Model 3, which was approximately 20 times the regression performance of the model using strength (Model 1 and 2). Ensemble learning is effective in reducing model bias and variance [38]. Thus, Model 4 was improved by preventing overfitting to near zero in Model 3. We revealed that the overall contribution of medical cooperation factors to the duration of hospitalization in each patient with femoral neck fracture was about 20%. This result indicates that the remaining variation is explained by factors other than medical cooperation. We show six possible factors in Eq (1): patient-specific factors *P*, health factors *H*, environment factors *E*, social service factors *S*, medical care factors *M*, medical cooperation factors *C*. We considered that there is the remaining

variation not explained by medical cooperation because these factors differed in each case. In Model 4, the accuracy of fitting $D^{in}$ was low in the domain where $D^{in}$ was large. This suggests that patients and the quality of medical care are provided at each medical provider contribute more than factors related to medical cooperation in prolonged hospitalization.

The community analysis results showed that each community prominently contained nodes in the same medical administration area. This result is consistent with the research findings on the structure of medical provider networks in China [39]. The geometric mean of strength $\mu_G(s)$ varied among communities, indicating differences in the level of medical cooperation among communities. Fig 11 shows the relationship between each network feature and $\mu_G(s)$ to explain this difference. The closer the distance between nodes, the more horizontally distributed in the structure and the higher the level of medical cooperation. In contrast, the centralized structure may lead to bottlenecks, and prolonged hospital stays when patients are centered in a hub medical provider.

Patients admitted to medical providers with a high level of medical cooperation had a shorter duration of hospital stay. The regional medical inter-institutional cooperation inferred from the medical provider network explains approximately 20% of the variation in the duration of hospitalization. The duration is an evaluation of the healthcare quality. We expect to develop a medical cooperation measure based on the network features composed of interpretable network features, which has the same level of explanatory ability as the feature representation using node2vec. This measure could be used to assess regional medical inter-institutional cooperation as a guide for introducing medical policies. We also consider the optimization of the healthcare system using this measure. In addition, although this study focused on femoral neck fractures, it is also necessary to investigate the relationship between medical cooperation and the quality of healthcare for other diseases. It is possible to clarify the contribution of medical cooperation with each disease using the method described in this study. This could identify diseases in which medical cooperation contributes to the quality of healthcare provision.

## Summary

It is necessary to construct an efficient healthcare system and provide good quality healthcare for the aging population worldwide. The networks among medical providers and physicians constructed by patient claims data have been considered to represent medical cooperation. Previous studies have revealed the relationship between the quality of healthcare provision and network features. In contrast, patterns of multiple medical providers used by patients have not been taken into account. They used only general network features to explain the quality of healthcare provision. The overall contribution to the quality of healthcare provision from the information extracted from medical networks is unknown. In addition, it is important to incorporate the usage patterns of medical providers in the Japanese health care system. This study aimed to examine the overall contribution to the quality of health care provision from medical inter-institutional cooperation, using the information on medical cooperation in the medical provider network in Japan.

We conclude that the regional medical inter-institutional cooperation represented by the medical provider network is an important factor in shortening the duration of hospital stay. We examined a model that uses the node2vec feature representation as input variables and ensemble learning as a regression model to improve the explanatory ability and found that the overall contribution of medical inter-institutional cooperation to the quality of healthcare provision at case level was approximately 20%. Other factors, such as the patient condition and care by the individual hospital, explained the remaining variation in the quality of healthcare. Medical inter-institutional cooperation still plays a significant role in providing high-quality

healthcare. In addition, the results for community analysis suggested that the horizontally distributed network structure is effective for medical inter-institutional cooperation.

## Supporting information

**S1 Fig. Distribution of inpatient and outpatient intervals.** (a) Cumulative distribution function (CDF) of inpatient intervals. (b) Probability distribution function (PDF) of inpatient intervals. (c)Cumulative distribution function (CDF) of outpatient intervals. (d)Probability distribution function (PDF) of inpatient interval. The distribution of inpatient intervals followed an exponential distribution ($\mu = 126$), except for the zero-day interval. The outpatient interval has a cyclic variation with a peak every 7 days. The inpatient interval distribution included approximately 60% of the inpatient interval at $\tau_1 = 10$ days. The outpatient interval distribution included approximately 80% of the outpatient interval at $\tau_2 = 35$ days.
(TIF)

**S2 Fig. Duration of hospital stay $D^{in}$ vs. mean and geometric mean of strength.** (a) shows mean $\mu(s)$. (b) shows the geometric mean $\mu_G(s)$. The geometric mean has a better explanatory ability when both $R^2$ values are compared.
(TIF)

**S3 Fig. Pre-analysis for determining hyperparameters of node2vec.** (a) number of walkers per node $t$; (b) Length of random walk $l$; (c) Window size $w$; and (d) Dimension $d$. We set the default parameter values as $(t, l, w, d, p, q) = (5, 25, 10, 70, 1, 1)$, and extracted feature representations by adjusting $t$, $l$, $w$, and $d$ by one parameter each. We performed a regression analysis using the extracted $\boldsymbol{v}$. We calculated the root mean squared error (RMSE) using $\hat{D}^{in}$ obtained from regression analysis. We performed 20 iterations with five-fold cross-validation and used the average value of the RMSE as the evaluation value of the regression performance.
(TIF)

**S4 Fig. Surface plot of the grid search of hyperparameters $p$ and $q$ of node2vec $p \in \{4, 8, 16, 32, 64, 128\}$ and $q \in \{1/128, 1/64, 1/32, 1/16, 1/8, 1/4\}$.** We evaluated the regression performance with changes in parameters $p$ and $q$ using the RMSE. The figure shows that the RMSE approaches the plane when $p$ increases, and $q$ decreases. This means that there are a large number of local minima, which makes it difficult to find a unique optimum value.
(TIF)

## Author Contributions

**Conceptualization:** Yuichi Ikeda, Yuichi Imanaka.

**Data curation:** Yu Ohki, Susumu Kunisawa.

**Formal analysis:** Yu Ohki.

**Funding acquisition:** Yuichi Imanaka.

**Investigation:** Yu Ohki, Yuichi Ikeda.

**Methodology:** Yu Ohki, Yuichi Ikeda.

**Project administration:** Yuichi Ikeda.

**Resources:** Yu Ohki, Susumu Kunisawa.

**Software:** Yu Ohki.

**Supervision:** Yuichi Ikeda.

**Validation:** Yu Ohki.

**Visualization:** Yu Ohki.

**Writing – original draft:** Yu Ohki.

**Writing – review & editing:** Yuichi Ikeda, Susumu Kunisawa, Yuichi Imanaka.

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
