## [Decision Letter · Decision Letter 0]

27 Jun 2022

PONE-D-22-07692Regional medical inter-institutional cooperation in medical provider network constructed using patient claims data from JapanPLOS ONE

Dear Dr. Ohki,

Thank you for submitting your manuscript to PLOS ONE. After careful consideration, we feel that it has merit but does not fully meet PLOS ONE’s publication criteria as it currently stands. Therefore, we invite you to submit a revised version of the manuscript that addresses the points raised during the review process.

We look forward to receiving your revised manuscript.

Kind regards,

Hocine Cherifi

Academic Editor

PLOS ONE

Journal Requirements:

“This work was supported by JSPS KAKENHI (Grant Number: JP19H01075) from the Japan Society for the Promotion of Science and Health and Labour Sciences Research Grant from the Ministry of Health, Labour and Welfare, Japan (Grant Number: 21IA1005 and 21FA1012). YO thanks the Kyoto University Science and Technology Innovation Creation Fellowship. Y. Ikeda and YO would also like to acknowledge Ripple, which is providing financial support through its University Blockchain Research

Initiative.”

Additional Editor Comments :

This paper studies medical cooperation efficiency through medical provider and physician networks based on patient claim data. They are starting with a bipartite network of a patient using medical providers. They build the projected networks. They extract network features to predict the quality of healthcare. They computed the duration of hospital stay of patients in each case of surgery related to the femoral neck 633 fracture. They also use community structure analysis to investigate the relationship between the level of medical cooperation and network structure. They show that the regional medical inter-institutional collaboration represented by the medical provider network is an essential factor in shortening the duration of hospital stay. The paper is well written. Experiments and results are sound. The methodology and the findings are of great interest to the scientific community. Therefore, I recommend publication after taking care of the minor revisions suggested by the reviewer.

Reviewers' comments:

Reviewer's Responses to Questions

**Comments to the Author**

1. Is the manuscript technically sound, and do the data support the conclusions?

Reviewer #1: Yes

2. Has the statistical analysis been performed appropriately and rigorously? 

Reviewer #1: Yes

3. Have the authors made all data underlying the findings in their manuscript fully available?

Reviewer #1: Yes

4. Is the manuscript presented in an intelligible fashion and written in standard English?

Reviewer #1: Yes

5. Review Comments to the Author

Reviewer #1: Overall, this paper is excellent.

This paper uses standard network science notations and so is generally easy to read and figure out.

The work is strong, and the results are quite interesting.

I have only small comments, listed below.

Abstract:

In the abstract and intro you mention 4 types of models. It would be useful to simply list the 4 types, maybe even as 1, 2, 3, 4, like what you did with the study focus. Or something like: “Models 1 and 2 use node strength and linear regression, with Model 2 also incorporating patient age as an input. Models 3 and 4 use feature representation by node2vec with linear regression and regression tree ensemble, a machine learning method.”

Also, in both abstract and intro you use the word “strength” (or a “stronger medical provider”)

to describe medical providers, and it is not totally clear what it means in that context (it hasn’t been defined yet – l assume you mean it as defined on line 170). It might be better to say “The results showed that medical providers with higher levels of cooperation reduce the duration of hospital stay.” (as you do on lines 414-415)

Introduction:

Very small language errors. For example in the first paragraph, “establish healthcare systems” and “services to address this situation.”

The organization section is confusion, referring to “section 2,” although the sections aren’t numbered.

Line 108: “recovery phase hospitals”

Line 125: “the following function”

Data and Methods:

I like the fact that you have used cosine similarity to normalize size of providers.

Line 181: Might be useful to define assortativity.

Line 233: This is my favorite paragraph in the paper. You have done an excellent job of explaining things.

Overall I find this section to be well done. Good job.

Results

Line 399 Din is typeset wrong.

Line 492 talks about 7 communities, but table 6 only shows 6.

Discussion

Line 522 I don’t think the models need to be repeated.

This section is very well done.

Summary

This section is highly repetitive, and you have already said the important things in the discussion. I don’t think you need the second paragraph at all, and the 3rd paragraph is mostly repetition.

6. PLOS authors have the option to publish the peer review history of their article (what does this mean?). If published, this will include your full peer review and any attached files.

Reviewer #1: No

---

## [Author Response · Author response to Decision Letter 0]

19 Jul 2022

Response to Reviewers 2022/07/12

We would like to thank the reviewers for their helpful comments. After serious consideration of all the comments, the paper has been revised as follows.

Revision

Abstract:

In the abstract and intro you mention 4 types of models. It would be useful to simply list the 4 types, maybe even as 1, 2, 3, 4, like what you did with the study focus. Or something like: “Models 1 and 2 use node strength and linear regression, with Model 2 also incorporating patient age as an input. Models 3 and 4 use feature representation by node2vec with linear regression and regression tree ensemble, a machine learning method.”

Response to the comment: We described the four models we focused on in this research following your suggestion: "We considered four types of models. Models 1 and 2 use node strength and linear regression, with Model 2 also incorporating patient age as an input. Models 3 and 4 use feature representation by node2vec with linear regression and regression tree ensemble, a machine learning method."

Also, in both abstract and intro you use the word “strength” (or a “stronger medical provider”)

to describe medical providers, and it is not totally clear what it means in that context (it hasn’t been defined yet – l assume you mean it as defined on line 170). It might be better to say “The results showed that medical providers with higher levels of cooperation reduce the duration of hospital stay.” (as you do on lines 414-415)

Response to the comment: We changed the sentence following your suggestions: "The results showed that medical providers with higher levels of cooperation reduce the duration of hospital stay.”

Introduction:

Very small language errors. For example in the first paragraph, “establish healthcare systems” and “services to address this situation.”

Response to the comment: We changed the sentence to "establish healthcare systems" and "services to address this situation."

The organization section is confusion, referring to “section 2,” although the sections aren’t numbered.

Response to the comment: We changed the paragraph to "The remainder of this paper is organized as follows. The following section describes the role of medical cooperation among healthcare providers for a femoral neck fracture. We then describe the data, data analysis method, statistical model, and community analysis in the "Data and Method" section. The "Results" section presents the relationship between the features of medical providers and the duration of hospital stay based on the constructed network. Using a community analysis, we also examine the relationship between the network structure and medical cooperation. The "Discussion" section discusses the results, and the "Summary" section summarizes and concludes this paper."

Line 108: “recovery phase hospitals”

Response to the comment: We changed the sentence to "recovery phase hospitals."

Line 125: “the following function”

Response to the comment: We used the term "functional" because each component of Eq. (1) is a function and f represents a higher-order function. Thus, "the following functional" is correct.

Data and Methods:

I like the fact that you have used cosine similarity to normalize size of providers.

Line 181: Might be useful to define assortativity.

Response to the comment: We changed the sentence to "Assortativity r is the Pearson correlation coefficient of the degree of the nodes at both ends of the edge, r=Σ_ij (k_i k_j-〈k〉^2)/〖Σ_i (k_i-〈k〉)〗^2."

Line 233: This is my favorite paragraph in the paper. You have done an excellent job of explaining things.

Overall I find this section to be well done. Good job.

Results

Line 399 Din is typeset wrong.

Response to the comment: We revised it to "D^in."

Line 492 talks about 7 communities, but table 6 only shows 6.

Response to the comment: We revised table 6 to add a "Community 7" column.

Discussion

Line 522 I don’t think the models need to be repeated.

Response to the comment: We removed line 522 to line 531.

This section is very well done.

Summary

This section is highly repetitive, and you have already said the important things in the discussion. I don’t think you need the second paragraph at all, and the 3rd paragraph is mostly repetition.

Response to the comment: We removed the 2nd paragraph. We merged the 3rd paragraph and 4th paragraph: " We conclude that the regional medical inter-institutional cooperation represented by the medical provider network is an important factor in shortening the duration of hospital stay. We examined a model that uses the node2vec feature representation as input variables and ensemble learning as a regression model to improve the explanatory ability and found that the overall contribution of medical inter-institutional cooperation to the quality of healthcare provision at case level was approximately 20%. Other factors, such as the patient condition and care by the individual hospital, explained the remaining variation in the quality of healthcare. Medical inter-institutional cooperation still plays a significant role in providing high-quality healthcare. In addition, the results for community analysis suggested that the horizontally distributed network structure is effective for medical inter-institutional cooperation."

---

## [Editor Report · Decision Letter 1]

3 Aug 2022

Regional medical inter-institutional cooperation in medical provider network constructed using patient claims data from Japan

PONE-D-22-07692R1

Dear Dr. Ohki,

We’re pleased to inform you that your manuscript has been judged scientifically suitable for publication and will be formally accepted for publication once it meets all outstanding technical requirements.

Kind regards,

Hocine Cherifi

Academic Editor

PLOS ONE
---

## [Editor Report · Acceptance letter]

12 Aug 2022

PONE-D-22-07692R1 

Regional medical inter-institutional cooperation in medical provider network constructed using patient claims data from Japan 

Dear Dr. Ohki:

I'm pleased to inform you that your manuscript has been deemed suitable for publication in PLOS ONE. Congratulations! Your manuscript is now with our production department. 

Kind regards, 

on behalf of

Professor Hocine Cherifi 

Academic Editor

PLOS ONE